# Evaluation of Electric Muscle Stimulation Method for Haptic Augmented Reality [note 1]

**DOI:** 10.3390/s23041796

**Published:** 2023-02-05

**Authors:** Takaya Ishimaru, Satoshi Saga

**Affiliations:** 1Department of Advanced Industrial Science, Kumamoto University, 2-39-1 Kurokami, Kumamoto City 860-8555, Japan; 2Faculty of Advanced Science and Technology, Kumamoto University, 2-39-1 Kurokami, Kumamoto City 860-8555, Japan

**Keywords:** electric muscle stimulation, haptic augmented reality, waveforms

## Abstract

Currently, visual Augmented Reality (AR) technology is widespread among the public. Similarly, haptic AR technology is also widely practiced in the academic field. However, conventional haptic AR devices are not suitable for interacting with real objects. These devices are often held by the users, and they contact the real object via the devices. Thus, they prevent direct contact between the user and real objects. To solve this problem, we proposed employing Electrical Muscle Stimulation (EMS) technology. EMS technology does not interfere with the interaction between the user and the real object, and the user can wear the device. First, we examined proper stimulus waveforms for EMS, in addition to pulse waveforms. At the same time, we examined the appropriate frequency and pulse width. The waveforms that we used this time were a sawtooth wave, a reverse sawtooth wave, and a sine wave. Second, to clarify the characteristic of the force presented by the EMS, we measured the relationship between the input voltage and the force presented and obtained the point of subjective equality using the constant method. Subsequently, we presented the bump sensation using EMS to the participants and verified its effectiveness by comparing it with the existing methods.

## 1. Introduction

In recent years, there has been much research on Augmented Reality (AR) technology that expands the perceptual ability of humans. Accessible AR technology is mainly visual AR. For example, by superimposing the information of furniture and characters on the visual information of the real world, we can see the combined information.

There also exist several types of research on AR technology in the haptics. Haptic AR technology based on force feedback devices is often used to improve operability. For example, some haptic AR systems augment physical properties, such as stickiness, weight, and friction, by interactively controlling force [1]. In addition, there is an AR system that improves operability by limiting the user’s movement [2].

However, existing haptic AR systems have problems. One is that it is necessary to install a vibrator and a force presentation function on the device. However, the required number becomes large when the user’s interaction occurs with a lot of real objects. Another problem is that when the user wears or grips the device, the device consequently often exists between the user and the real object, which prevents direct interaction between the user and the real object.

In this paper, we proposed a method that users can wear, and that can give a force sensation without preventing direct contact with real objects. Furthermore, we examined the proper stimulus waveforms and frequency for EMS, other than pulse waveforms. Second, we clarified the characteristic of the force presented by the EMS. After that, we presented the sensation of bumps using EMS to the participants and verified the effectiveness by comparing it with the existing method.

## 2. Background

Haptic AR technology based on force feedback devices is often used to improve operability. In conventional research, the exoskeleton-type power assist device, such as HARDIMAN [3], developed in 1965, is a haptic AR in the sense of reducing the weight of the real object. Furthermore, the concept of haptic AR has appeared in research using force-feedback devices [4]. Some haptic AR systems augment physical properties, such as stickiness, weight, and friction, by interactively controlling force [1]. For example, if friction could be increased by haptic augmentation, it would become possible to grip things that would otherwise be slippery and difficult to grab more stably than usual. In addition, there is an AR system that improves operability by limiting the user’s movement [2]. Specifically, in carving skills, it is possible to carve as designed by creating an inaccessible area with sculpture.

In addition, there is an AR system that improves operability by adding tactile information using a vibrator [5]. For example, the system can present a click feeling by employing vibration when a touch event occurs. This feeling of clicking enhances the sense of touch so that the user can perceive the contact towards the screen for sure, and the operability is improved. These examples mainly change the haptic information by adding virtual haptic information to the haptic information of the real object that the user touches. In addition, there is also an AR system that applies sensory substitution technology. The system modifies and presents sensory information, such as visual information, as haptic information. This AR system is especially important in the welfare field and presents haptic sensations as substitute information for users who have some disabilities in physical perception functions. This haptic AR information is presented by superimposing haptic information on real sense according to the user’s subjective actions, such as looking over real spaces. For example, many attempts have been made to present the visual information captured by the camera as haptic information [6]. In conventional haptic AR systems [1,2,6], the haptic information of an object is changed or complemented to enhance human haptic ability, improve operability, and substitute sense.

However, these haptic AR systems have a problem. One is that it is necessary to install a vibrator and a force presentation function on the device. For example, a vibration touch panel or force-displaying pen device requires actuators on the device side to present a haptic sensation. This is effective when performing haptic AR using a specific device; however, the required number becomes large when the user’s interaction occurs with a lot of real objects. Now that head-mounted displays are becoming popular, it is necessary to interact with many real objects simultaneously, and it is not realistic to add tactile devices to all of them.

Another problem is that when the user wears or grips the device, the device consequently often exists between the user and the real object, which prevents direct interaction between the user and the real object. For example, when holding a device, such as a controller, the user cannot touch the object directly. Under this condition, it is difficult for the user to perceive haptic AR sensations because the user cannot obtain the haptic information of a real object. Although bracelet-type devices or nail-mounted devices do not prevent contact with an object by using a vibrator, the vibrator is not good at presenting force sensation and can only present as strong or weak. Therefore, we proposed a method that users can wear, and that can give a force sensation without preventing direct contact with real objects. To realize the concept, we focused on Electrical Muscle Stimulation (EMS).

When a human moves their body, the brain sends commands to the peripheral nerves. When the command arrives, the nerves are excited, and this excitement is transmitted to the muscles, causing the muscles to contract and move each part of the body. Electrical Muscle Stimulation (EMS) uses this mechanism. External electrical stimulation induces excitation of peripheral nerves, causes muscle contraction, and presents an external force to the user [7]. Initially, EMS technology was used primarily in rehabilitation, such as moving the paralyzed limb [8] and helping to walk [9]. The technology is called Functional Electrical Stimulation (FES) when used to drive a paralyzed limb. “Possessed hand” [10] has led to some studies that use EMS as an interface. For example, there is a study that augments the feeling of hitting a ball in a simple tennis game by contracting the wrist muscles [11] and a study that presents a wall in a VR space [12]. There are many other studies to improve the reality of haptics [13,14,15]. Many studies use EMS to support people [16,17], such as teaching how to play musical instruments [18] and training [19]. Most HCI research about EMS with AR only treats the augmentation of vision. However, the users do not get haptic information from real objects but only get virtual force via EMS [20]. Other studies have also used EMS as a means of communication [21,22,23]. Our research concept is to augment the virtual force on the real force at the user’s finger directly. To realize the concept, we evaluated EMS-based methods.

Our research purpose is to clarify how to design the input signals for the EMS method. To evaluate the usefulness of this method, we examined the proper stimulus waveforms for EMS, other than pulse waveforms. At the same time, we examined the appropriate frequency and pulse width. The waveforms that we used this time were a sawtooth wave, a reverse sawtooth wave, and a sine wave. Second, to clarify the characteristic of the force presented by the EMS, we measured the relationship between the input voltage and the force presented and obtained the point of subjective equality and the just noticeable difference using the constant method. After that, we presented the sensation of bumps using EMS to the participants and verified the effectiveness by comparing it with the existing method. Part of this paper was published at the Haptics Symposium 2020 [24] and the IEEE/SICE International Symposium on System Integration 2022 [25].

In the experiment, we explained to participants all the potential risks associated with the study, and they gave their written informed consent to participate in the study. The Ethics Committee of the Graduate School of Advanced Sciences, Kumamoto University, allowed this investigation with receipt number R1-2.

## 3. Consideration of Waveforms for EMS

This section describes the stimulus waveforms we evaluated in the experiment. Here, we evaluated new stimulus waveforms other than pulse waveforms. At the same time, we also evaluated the appropriate frequency and pulse width. The waveforms that we used this time were a sawtooth wave, a reverse sawtooth wave, and a sine wave. We used two types of sawtooth waves. Figure 1 shows the waveforms that we used. We created these with safety in mind, referring to the study of design guidelines [26].

For normal EMS, we used the pulse wave with a width of 0.6 ms. Here, we aimed to suppress discomfort and electrical feelings and designed the following waveforms by slowing the change in voltage. The sawtooth waveform has a gentle slope at the rising edge, the reverse sawtooth wave has a gentle slope at the falling edge, and the sine wave has gentle slopes for both edges. We set these waveforms whose time-integrated area to be the same as the pulse wave with a width of 0.6 ms. If skin resistance could be assumed to be the same between each condition, the waveforms would give the same charge to the skin. Specifically, we set sawtooth wave #1 as multiplying the stimulation time and maximum amplitude by $\sqrt{2}$ compared to the pulse wave, and sawtooth wave #2 as keeping the maximum amplitude and doubling the stimulation time compared to the pulse wave. We also made the reverse sawtooth waves #1 and #2 have a similar design. For the sine wave, we kept the maximum amplitude and set the stimulation time to twice the pulse wave.

In addition to these five waveforms, we used pulse waves with five different frequencies and five different pulse widths. The waves of the five different frequencies were 20, 35, 50, 65, and 80 Hz, whose pulse widths are all 0.6 ms. The five waves of different pulse widths were 0.2, 0.4, 0.6, 0.8, and 1.0 ms, whose frequencies were all 50 Hz. There were ten conditions, though, the wave conditions with 0.6 ms of pulse width and 50 Hz of frequency overlapped, so we used a total of nine types of pulse waves. We determined these frequencies and pulse waves from preliminary experiments and previous studies [25]. Therefore, we conducted experiments using a total of 14 types of waveforms: 9 types of pulse waves, 2 types of sawtooth waves, 2 types of reverse sawtooth waves, and 1 type of sine wave.

### 3.1. Electrode Positioning for Electrical Muscle Stimulation

This section describes the force presentation method using EMS proposed in this paper. Usually, nerves travel deep into the body, so stimulating nerves from the skin’s surface is not easy. However, the motor point (MP) is the part that responds the most to electrical stimulation when stimulating muscles using surface electrodes. In this study, we stimulated the MP of the extensor muscle using EMS, induced muscle contraction, and presented a force to the participants’ finger.

The MP is the part where this nerve enters the superficial muscle from the deep part, and we can efficiently induce muscle contraction by placing electrodes on this MP [27]. Generally, it is not easy to estimate the position of the electrode [28]. However, in this paper, we proposed a method to estimate the MP position. Our method dynamically slid the electrode while measuring the presentation force and determined that the place where muscle contraction could induce more efficiently than other parts was the MP. Specifically, we determined that where the participant exerts a force of 0.3 N with no subjective effort as the MP. By placing electrodes on the MP, we could efficiently cause muscle contraction and, at the same time, suppress the tingling electric sensation peculiar to EMS.

### 3.2. Evaluation Experiment of Input Waveforms

In this section, we explain the experiments we conducted to examine the appropriate frequency and pulse width between the 14 types of waves to suppress discomfort and electrical feelings. We show an overview of the measurement system in Figure 2.

There were seven participants in the experiment, all men, with an average age of 23.1 years and a standard deviation of 0.6. We used 14 types of waveform that we described in the previous section. As a single trial, after stimulating with a waveform for 5 s, participants evaluated the following three sensations; electrical sensation, discomfort, and pain on a 5-stage Likert scale. The participants performed three trials for each type of waveform. Therefore, each participant conducted a total of 42 trials with 14 type × 3 trials. Therefore, we obtained a total of 294 responses. We recorded the voltage and the presentation force at the same time during the experiment.

As we explained at the beginning of the section, we used 14 patterns of waveform and pulse width for input stimuli. In addition, we had to determine the voltage amplitude of the input stimuli for each participant individually. This was because our aim was to unify the presentation force, not the voltage. Even if the voltage was constant, there was a difference in the presentation force due to individual differences in the electrical characteristics of the participants. Then, we adjusted the amplitude so that the presentation force became 0.3 N in the case of a pulse wave with 0.6 ms of pulse width and 50 Hz of frequency to the extensor muscle of the participants’ index finger.

Next, we explain the equipment that we used in this experiment. We used a high-voltage electrical stimulator developed by Kajimoto Laboratory [29]. This electrical stimulation device can control waveforms with 12 bits using a microcomputer called mbed. The maximum output current is 10 mA, and the rated output voltage is 300 V. To measure the stimulation waveform, we used Tektronix’s TDS 2004B digital oscilloscope, and to measure the presentation force, we used Imada’s ZTA-50N. The electrode to stimulate the user was the ECG electrode, Medico MSGLT-04, manufactured by AS ONE Corporation (Tokyo, Japan).

### 3.3. Results of Waveform Experiment

We will now show the results of the experiment. Figure 3a shows the result of the sensation caused by the change in frequency in the pulse wave. Figure 3b shows the result of the sensation due to the change in pulse width. Figure 3c shows the result of the sensation caused by the change in the waveform.

In Figure 3a–c, the orange lines show the answer of “discomfort”, the blue lines show “pain”, and the green lines show “electric feeling”. The vertical axes are the evaluation results on a 5-stage Likert scale. The maximum value is 5 and the minimum value is 1, and the larger the value, the greater the sensation. Each horizontal axis represents the frequencies, pulse widths, and waveforms of the input stimuli. Pairwise comparisons between each condition after one-way repeated measures ANOVA were made using Scheffe’s F-test.

First, we discuss the frequency effect (Figure 3a). We found no statistically significant differences as a result of the change in frequency, but all sensations become stronger as the frequency increases.

Next, we discuss the pulse width effect (Figure 3b). We found statistically significant differences at the significance level of 5% between most conditions. The “n. s.” in Figure 3b shows the combinations that have no statistically significant differences, and all the others have statistically significant differences. The “n. s.” pairs are all adjacent waves whose width difference is 0.2 ms. Therefore, we found that the wave pairs between the difference over 0.4 ms have some statistically significant differences. We also confirmed that increasing the pulse width increases the three sensations.

Finally, we discuss Figure 3c. There were several statistically significant differences between several pairs, though our motivation was to compare the original pulse wave with the others, and we showed only combinations that were statistically significant differences from pulse waves with the “*”s in the figure. We found that between pulse wave and sawtooth wave #2 and pulse wave and reverse sawtooth wave #2, there were statistically significant differences at the significance level of 5%. Sawtooth wave #2 and reverse sawtooth wave #2 had the same maximum amplitude and twice the pulse width compared to the pulse wave. Thus, the voltage changes slowly. Therefore, we considered that a gentle slope of the voltage change in sawtooth #2 and reverse sawtooth #2 reduced discomfort, pain, and electric feeling.

### 3.4. Result of Voltage and Presentation Force

Next, we show the measurement results of the presentation force for each frequency, pulse width, and waveform condition. Figure 4a shows the results of the frequency change. Figure 4b shows the results of the pulse width change. Figure 4c shows the results for each stimulus waveform.

The red line indicates the presentation force for each input stimulus. The vertical axis indicates the force presented to the index finger. The error bar indicates the standard deviation. First, we discuss the frequency (Figure 4a). In terms of force, we could see that the higher the frequency, the greater the presentation force. We considered this because the amount of current flowing per second increases as the number of stimuli increases. The green regression line also shows the result. As you can see, there was a linearity between the input frequency and the presentation force.

Next, we discussed pulse width (Figure 4b). In terms of the force, we could confirm that the presentation force tends to increase as the pulse width increases. We considered this because the amount of current that flows per time increases as the pulse width increases. The green regression line also shows the result. There was a linearity between the input pulse width and the presentation force. In other words, when the pulse width became small, the amount of current flowing per time became small. Thus, we confirmed that the presentation force was clearly reduced.

Then we discuss the stimulus waveform (Figure 4c). We confirmed that a large force was measured from one specific participant, which increased the average value. The interview after the experiment revealed that the participant may have unintentionally applied force to the finger in the experiment. Therefore, we excluded the results of the participant as outliers and found that reverse sawtooth wave #1 had almost the same result as sawtooth wave #1.

Pairwise comparisons between each condition after one-way repeated measures ANOVA were made using Scheffe’s F-test. There were several statistically significant differences between several pairs, though our motivation was to compare the original pulse wave with the others. However, there were no statistically significant pairs with the pulse wave. Although there were no statistically significant differences, the reverse sawtooth wave #2 has a low presentation force on average. This was a waveform in which the voltage gradually decreased after the first stimulation with the maximum value. However, when a current flowed into the human body, the voltage changed slowly. Therefore, perhaps the voltage had not risen completely. Furthermore, even though we presented a similar voltage in sawtooth wave #1, we measured the presentation force almost the same as the pulse wave on average. We considered this because sawtooth wave #2 gradually increases the voltage, and the time to reach the maximum voltage was only the last moment. Similarly, the reverse sawtooth wave has the time stimulated by the maximum voltage only at the first moment.

On the basis of the discussion above, we consider waveforms suitable for Electrical Muscle Stimulation. Ideally, we should present the force without any discomfort, pain, or electrical sensation. Therefore, we focused on the conditions under which these three sensations were low. Regarding pulse waves, pulse waves with a low frequency and a small pulse width seemed appropriate. However, if we reduce the frequency and pulse width, the presentation force will decrease. Therefore, we considered it appropriate that the pulse width was about 0.6 ms and the frequency was 50 Hz or higher.

Regarding waveforms other than the pulse wave, the undesired sensation was smaller in sawtooth wave 2 and reverse sawtooth wave 2. Still, the waveforms could not present enough force, so it is inappropriate. In other words, in the waveform we examined this time, a pulse wave with a pulse width of 0.6 ms and a frequency of 50–80 Hz was suitable for EMS because these conditions could present force while suppressing three feelings. On the other hand, other waveforms might be suitable depending on the purpose, such as increasing the presentation force or presenting the feeling of electricity. Here, we defined the 0.6 ms pulse wave with 50 Hz and amplitude with 1.0 as p(t). In the following section, we employed this wave as a fundamental waveform.

In addition, the linearity between the amount of current per time and the presentation force suggested it is possible to control the force. Using this linearity, we created an application in Section 5.

## 4. Evaluation Experiment of Force Sensation by EMS

This section investigated the point of subjective equality (PSE) of the EMS force and the physical force, which had not been clarified so far, and obtained an index of force strength when using EMS for haptic feedback. We calculated the PSE using the method of constant stimuli about the EMS force and the physical force (traction force by the motor). We obtained the PSE from a discrimination experiment through the method of constant stimuli. In addition, we investigate the just noticeable difference (JND) at the same time. In this experiment, we used a pulse wave with a 50 Hz frequency and a pulse width of 0.6 ms based on the results of Section 3.2.

### 4.1. The Method of Constant Stimuli

The method of constant stimuli is one of the basic measurement methods in the psychophysical measurement method [30]. Psychophysics is the science of examining the functional relationships between physical events and their corresponding psychological events. Therefore, the psychophysical measurement method is a procedure for giving some functional expression between them. In some cases, we ask qualitative judgments, such as whether a certain sensation is the same as a certain sensation. In some cases, we ask for quantitative decisions such as the difference between a certain sensation and a certain sensation. The former is a problem of detection or discrimination, and the latter is a problem of sensory scale.

When two stimuli $S_{c}$ and $S_{s}$ are presented, $S_{c}$ is the equivalent stimulus of $S_{s}$ when the sensory and perceptual characteristics of $S_{c}$ are subjectively equivalent to the standard $S_{s}$. The value of $S_{c}$, at this time, is PSE. In addition, $S_{s}$ is called a standard stimulus, and $S_{c}$ is called a comparative stimulus. $S_{s}$ must be kept constant. When we force participants to answer one of the two choices, which is larger, $S_{c}$ or $S_{s}$, the reaction is that $S_{c}$ is larger than $S_{s}$ and $S_{c}$ is smaller than $S_{s}$. The point where these ratios are 1:1 is the PSE.

We explain the specific method of obtaining PSE. The participants answer which of the two presented forces is larger, and we obtain the ratio of each comparative stimulus. After that, we convert the calculated ratio from the normal distribution table to the *z* value and perform linear interpolation using the least-squares method. This gives *a* and *b* of the following Equation (Equation 1).
(1)$$\begin{array}{cc} z & =a+b\times s \end{array}$$

Here, *z* represents the *z* value, and *s* represents the stimulus intensity. After that, we obtain PSE from the point where the ratio is 0.5. In other words, we obtain *s* from the point of z=0 in Equation (Equation 1). Further, the points at which the ratio of responses is 0.75 and 0.25 are the upper and lower discrimination thresholds, respectively. In other words, we can obtain *s* by substituting *z* = 0.67449, −0.67449, from which we can calculate the PSE.

### 4.2. Comparison between Induced Force by EMS and Physical Force

We used the method of constant stimuli to examine the PSE of the constant stimulus and the comparative stimulus. We did not change the constant stimulus in this experiment. We randomly changed the comparative stimulus. In this experiment, we used a 0.3 N force by EMS as a constant stimulus and 0.1, 0.2, 0.3, 0.4, 0.5, 0.6, 0.7, 0.8, 0.9, and 1.0 N physical force by motor as the comparative stimuli. We generated these comparative stimuli by winding the thread with a motor and pulling the finger attachment by the thread. From this experiment, we obtained the PSE between the force of the EMS and the physical force of a motor. At the same time, we calculated the JND.

We determined the comparative stimuli by a preliminary experiment with one of the authors considering whether he could reliably determine the stimulus to be larger or smaller than the constant stimulus.

Figure 5a shows a measurement of force by EMS, and Figure 5b shows a measurement of force by the motor. We instructed the participants not to put any strength on their fingers and to relax during the experiment. In the experiment, we presented the constant force of the EMS and the comparative force of the motor alternately and asked the participants to choose the larger.

Figure 6 shows the results of the experiment.

The vertical axis represents the probability that the comparative stimulus was greater than the constant force (0.3 N of EMS) determined by the participants (ratio of determination being greater). The horizontal axis represents the magnitude of the comparative stimulus. Each data point shows the probability that participants determined the comparative stimulus to be greater. The solid line shows the result of the interpolation by the cumulative distribution function. From the results, we got the PSE as 0.6 N. Here, the ratio of the comparative stimulus considered to be greater is 0.5. Furthermore, we got JND as 0.2 N. Therefore, participants could distinguish between physical force with a difference of 0.2 N. With EMS force, they would notice a difference of 0.1 N. When the EMS force was 0.3 N, participants felt that the force was twice as large as the actual force. In other words, the participants subjectively felt that the force of the EMS was twice as large as the force of the motor.

We considered that there were two reasons for this. One was that EMS stimulates the efferent nerve and the afferent nerve simultaneously and cause muscle contraction. Normally, when humans receive an external stimulus, the afferent nerve is stimulated, and this signal is transmitted to the brain. This time, EMS caused muscle contraction by stimulating the efferent nerve. In addition, we assumed that EMS causes the afferent nerve to be stimulated at the same time. In other words, since EMS stimulates both the efferent nerve and the afferent nerve simultaneously, humans subjectively feel about twice as large as regular stimulation that stimulates only the afferent nerve.

Second, it was possible that the peculiar electrical sensation based on EMS induced a greater force sensation. EMS usually generates a peculiar electrical sensation. From the comments after the experiment, 5 out of 10 participants said that they were a little concerned about the peculiar feeling of electricity. Thus, we considered that humans might feel a greater force due to the added peculiar electrical sensation.

As a result, the participants subjectively felt that the force was about twice as large as the force generated by EMS. They can recognize the difference if there is a difference of 0.2 N in physical force or 0.1 N in EMS force.

## 5. Haptic AR Display by EMS

In this section, we explain the haptic AR using EMS. After that, we explain the comparative experiments of haptic AR using existing methods and EMS and discuss the results.

### 5.1. Haptic Augmentation by EMS

In this section, we explain the haptic augmentation experiment. As shown in Figure 7, we presented AR-like virtual bump haptic stimulation by augmenting the shear force on the fingertip moving on the screen.

Saga et al. reported a virtual bump illusion using a lateral force representation device called the SPIDAR Mouse [31]. We used EMS to present the virtual bump illusion by presenting force instead of SPIDAR Mouse. Regarding the stimulus, referring to the previous study [31]; we changed the force according to the inclination of the virtual bumps that the user contacts. Next, we explain the formula with which we controlled the present force. We set the horizontal direction of the display as the *x*-axis, the shape of the bumps as s(x)=Asinx, the force as f(x), and the position of the finger as $x_{finger}$. We obtained the force using formula (Equation 3) because the force was proportional to the inclination of s(x).
(2)$$\begin{array}{cc} f(x) & \propto \frac{ds(x)}{dx} \end{array}$$
(3)$$\begin{array}{cc} \; & \propto A\cos x_{finger} \end{array}$$

We used this formula to present forces in both the positive and negative directions of the axis. For this purpose, we had to stimulate both the extensor and flexor muscles of the index finger. However, simultaneous stimulation of the extensor and flexor muscles was difficult because the flexor muscles of the fingers are located deep in the arm. Therefore, in this study, we only stimulated the extensor muscle and used only one directional force.

Thus, we added a constant value as an offset to generate a stimulus that had non-negative force. Therefore, the equation of the force is shown in Equation (Equation 4).
(4)$$\begin{array}{c} f(x)\propto A(\cos x+1) \end{array}$$

Using this offset, the user received an added constant force, even on the flat part of the bumps that have zero inclination. Using this force as a reference, we presented a smaller force than the reference as a negative slope and a larger force than the reference as a positive slope. This allowed the system to express bumps with only one directional force to the user.

Here, we set the maximum values for each participant determined by calibration as Max. Furthermore, as shown in Section 4.2, the pulse wave with 0.6 ms, p(t), was an appropriate waveform to display without discomfort. Thus, we multiplied the pulse wave p(t), and the displayed force f(x), and created the input voltage v(x,t) (Equation (Equation 5)).
(5)$$\begin{array}{cc} f(x) & =\frac{Max}{2}\cdot \left(\cos x+1\right) \end{array}$$
(6)$$\begin{array}{cc} v(x,t) & =p(t)\cdot f(x) \end{array}$$

By multiplying Max, Equation (Equation 5) changed from zero to the maximum value according to the gradient of the virtual bumps. In this study, we determined the maximum value for each participant/condition. In the next section, we compared this haptic AR displaying method using EMS with two other existing methods and verified whether it was practical to perform haptic AR using EMS.

### 5.2. Comparative Experiment about Haptic AR

This section verified the effectiveness of using EMS for haptic AR. We evaluated three methods, conventional vibrotactile stimulation [32], SPIDAR-based lateral force stimulation [31], and the proposed EMS-based stimulation with real bumps. We prepared three amplitudes of real bumps to evaluate the expressiveness of each method. By comparing them with the three stimuli of each method, we confirmed which method was possible to control the virtual amplitude of the stimuli. As low, medium, and high-amplitude real bumps, we prepared sinusoidal-shaped acrylic molds whose amplitudes were 1, 2, and 3 mm peak-to-peak, and their wavelengths were 30 mm.

As explained in the previous section, the stimulus changes according to the inclination of the virtual bumps. Specifically, we set the Max values of Equation (Equation 5) and determined the stimulus intensity f(x) according to the position of the finger $x_{finger}$. In this experiment, we set the maximum value Max for each stimulus based on the preliminary experimental results and the result of Figure 6 and adjusted each stimulus to match the real low, medium, and high bumps. Here, the following were the maximum values for each low, medium, and high conditions.

We set the Max value to feel as large as the other stimuli. For example, users felt that the maximum EMS value of 0.3 N was about the same as the shear force of 0.6 N and the vibration stimulus of a sine wave with an amplitude of 5.3 V.

**EMS1:** 0.30 N of EMS**EMS2:** 0.45 N of EMS**EMS3:** 0.60 N of EMS

**Vibration1:** Vibration stimulus by 100 Hz and 5.3 V of the sine wave**Vibration2:** Vibration stimulus by 100 Hz and 6.0 V of the sine wave**Vibration3:** Vibration stimulus by 100 Hz and 6.7 V of the sine wave

**SPIDAR1:** 0.60 N of lateral force**SPIDAR2:** 0.90 N of lateral force**SPIDAR3:** 1.2 N of lateral force

EMS was a method to present bump illusion by Electrical Muscle Stimulation, vibration was a method that uses a vibrator, and SPIDAR was a method that uses shear force using a SPIDAR mouse, similar to Saga et al. [31]. The vibrator method stimulus indicated the waveform input to the vibrator. We used a Force Reactor (Alps alpine, Co., Ltd., Tokyo, Japan) as the vibrator. The number in each condition name represents the intensity of the stimulus. We determined the difference in magnitude of these stimuli based on the JND obtained in Section 6. In one trial, we presented the participants with one virtual bump and one real bump, and they subjectively scored them according to their similarity using the magnitude estimation method. We used this score to evaluate the reproducibility of bumps. For real bumps, we prepared sinusoidal-shaped acrylic molds whose amplitudes were 1, 2, and 3 mm peak-to-peak and their wavelength was 30 mm. We randomly presented the virtual and real bumps to eliminate the order effect.

First, we evaluated the similarities between the methods. Reconstructing every result by methods, we created Figure 8.

Pairwise comparisons between each condition after one-way repeated measures ANOVA were performed using Scheffe’s F-test. From the result, the EMS method could reproduce the feeling of real bumps better than the other two methods.

Next, Table 1 shows the similarity ratio for each method to each real bump. There were three parameters: methods, stimulus intensities, and amplitudes of actual comparative bumps. The first row shows combinations of the methods and stimulus intensities. A numeric value follows each method name. The numeric value shows the stimulus intensity of each method. The first column shows the amplitudes of the actual bumps. Thus, a total of 27 results are shown.

Ideally, the diagonal elements of the three tables should be the highest score, and other cells should be lower than the diagonal elements. However, there were a small number of higher scores for diagonal elements. The SPIDAR table has higher scores in its diagonal elements and lower scores in peripheral cells. The EMS table has higher scores in its diagonal elements and lower in the left cells. The vibration table does not have a clear distribution of the higher/lower score.

This revealed that the SPIDAR method could reproduce the difference in intensities by changing the force. After the SPIDAR method, the EMS method could reproduce the difference in the lower intensities by changing the force. However, a higher intensity could not be clearly displayed. Finally, the vibration method could not reproduce the difference in intensities by changing the input voltage.

We summarize our considerations below. All the haptic AR methods could present the bumps with a significantly high degree of similarity. This shows the effectiveness of haptic AR for each method. In particular, from the result of the similarity score (Figure 8), we found that the EMS method is superior to the vibration and SPIDAR methods. From this, we found that we could present the force as we aimed and control the presentation force by the Equation (Equation 5). Furthermore, from Table 1, the SPIDAR method could reproduce the difference in intensities by changing the force. After the SPIDAR method, the EMS method could reproduce the difference in the lower intensities by changing the force. However, a higher intensity could not be clearly displayed. The vibration method could not reproduce the difference in intensities by changing the input voltage.

## 6. Conclusions

This paper aims to solve the problems of the existing haptic AR by using EMS, which users can wear and has the advantage of not interfering with the interaction between the user and the real object. First, we examined proper stimulus waveforms for EMS other than pulse waveforms. At the same time, we examined the appropriate frequency and pulse width. The waveforms that we used this time were a sawtooth wave, a reverse sawtooth wave, and a sine wave. Second, to clarify the EMS force’s characteristics, we obtained the Point of Subjective Equality (PSE) between the physical force and the EMS force using the method of constant stimuli. At the same time, we calculated the just noticeable difference (JND). After that, we presented the virtual bumps in haptic AR by EMS and verified the effectiveness by comparing them with existing methods.

In Section 1 and Section 2, we summarized the research background and proposed the use of EMS in haptic AR.

In Section 3, we examined the stimulation waveforms suitable for EMS, and found that using stimulation waveforms with a frequency of 35 to 50 Hz and a pulse width of 0.4 to 0.6 ms could present force without causing significant pain or discomfort. On the other hand, it turned out that it was difficult to reduce the electric feeling by adjusting the frequency and pulse width. We found the linearity between the amount of current per time and the presentation force, which suggested that we could control the force.

In Section 4, we investigated the force characteristics of EMS. We compared the force of the EMS and the physical force using the constant method. As a result, we found that users subjectively felt a physical force of 0.6 N compared to the EMS force of 0.3 N.

In Section 5, we performed a simple haptic AR using EMS and verified its effectiveness. We proposed a device that uses EMS to present the illusion of bumps by giving shear force. Using this device, we conducted a comparison experiment with existing methods. Finally, we found that EMS could present an illusion of bumps similar to real bumps better than vibration stimulation when we intended to give a bump illusion.

Through this paper, we investigated suitable stimulus waveforms for EMS for haptic presentation and the force characteristics of EMS. We found the linearity between the amount of current per time and the presentation force, which suggested that we could control the force. In addition, in this study, we presented tactile sensations on display with little texture information, so we could not obtain a statistically significant difference from haptic AR by SPIDAR. Still, it was superior to vibrators, which are commonly used as tactile presentation devices. In other words, by using EMS, we could realize a simple haptic AR without interfering with the direct interaction between the user and the object.

On the other hand, we could only realize the haptic AR on display. We have yet to verify whether we can realize haptic AR when real objects have complex texture information. In addition, we have yet to verify whether haptic AR with various real objects is possible. Taking into account these points, we were able to achieve part of the purpose of this research, but there are still things to verify. In the future, our aim is to realize haptic AR with complex texture information and to realize haptic AR with various real objects.

## Figures and Tables

**Figure 1 sensors-23-01796-f001:**
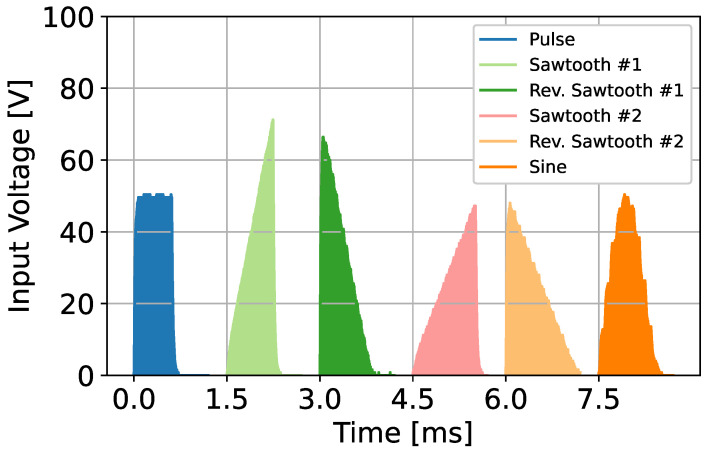
The stimulus waveform that we used in the experiment.

**Figure 2 sensors-23-01796-f002:**
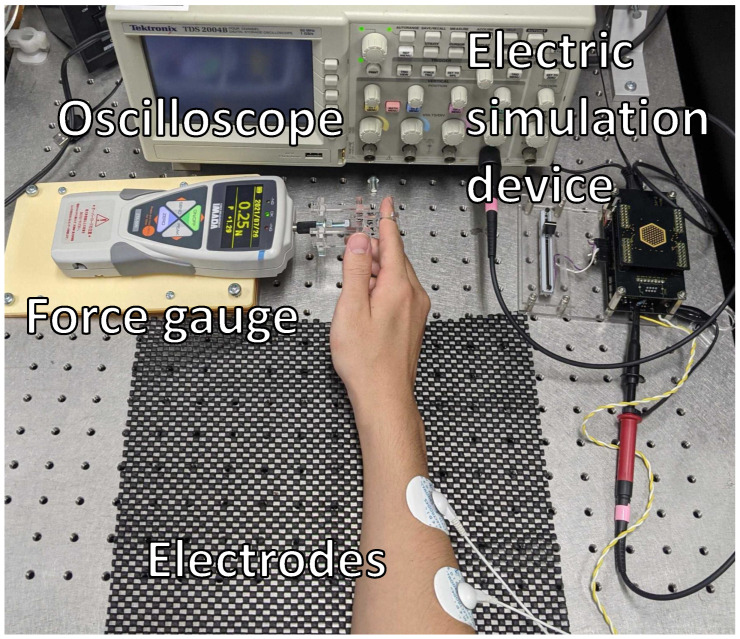
Overview of the voltage and presentation force measurement system.

**Figure 3 sensors-23-01796-f003:**
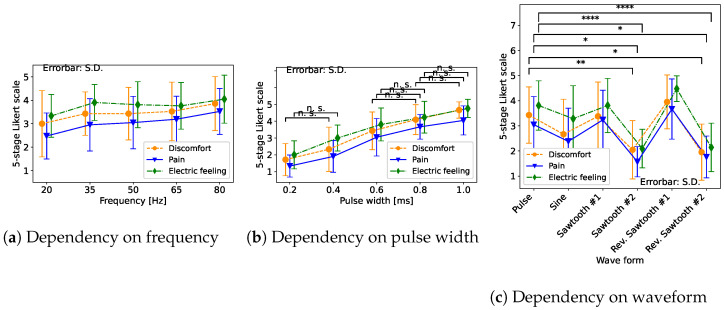
Changes in sensation (**a**) with frequencies; There are no statistically significant differences (**b**) with pulse width; we confirmed statistically significant differences except for the n. s. pair (**c**) with waveforms; this figure shows only combinations that are significantly different from pulse waves. *, ** and **** represent significance levels of 5%, 0.5%, and 0.005%.

**Figure 4 sensors-23-01796-f004:**
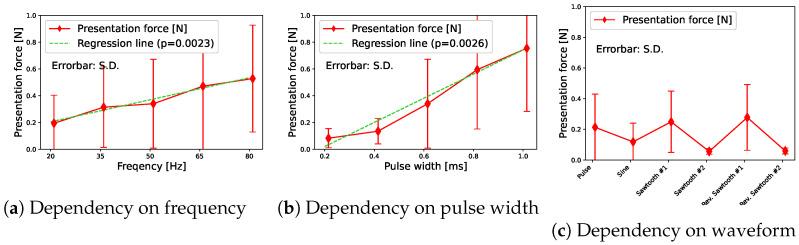
Presentation forces; (**a**) for each frequency; (**b**) for each pulse width; and (**c**) for each waveform. In (**c**), the outliers are excluded.

**Figure 5 sensors-23-01796-f005:**
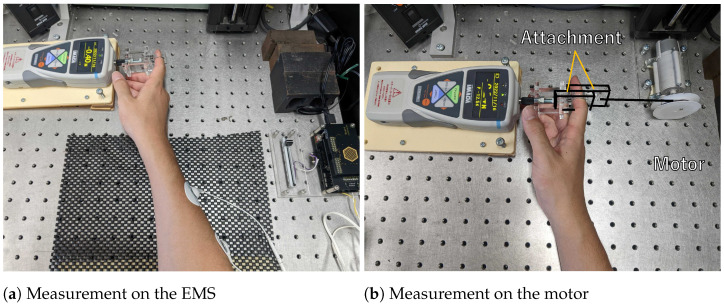
Measurement of forces and their PSE; (**a**) measurement of force induced by the EMS; (**b**) measurement of force generated by the motor.

**Figure 6 sensors-23-01796-f006:**
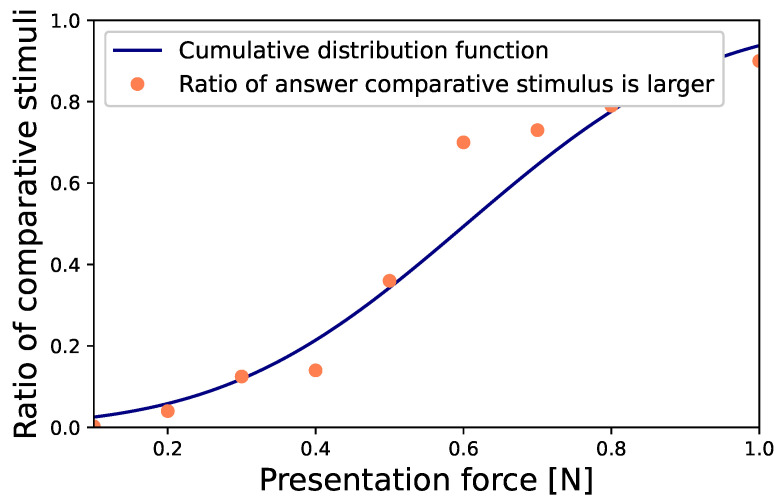
PSE between the force induced by the EMS and the motor.

**Figure 7 sensors-23-01796-f007:**
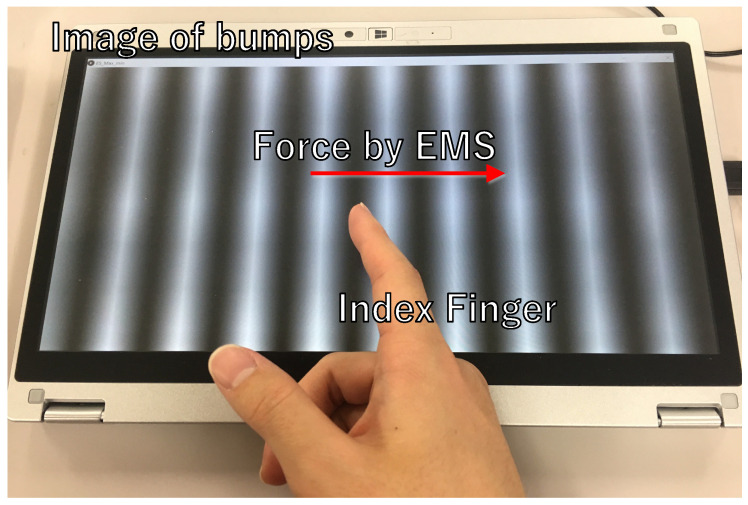
Haptic AR presentation system.

**Figure 8 sensors-23-01796-f008:**
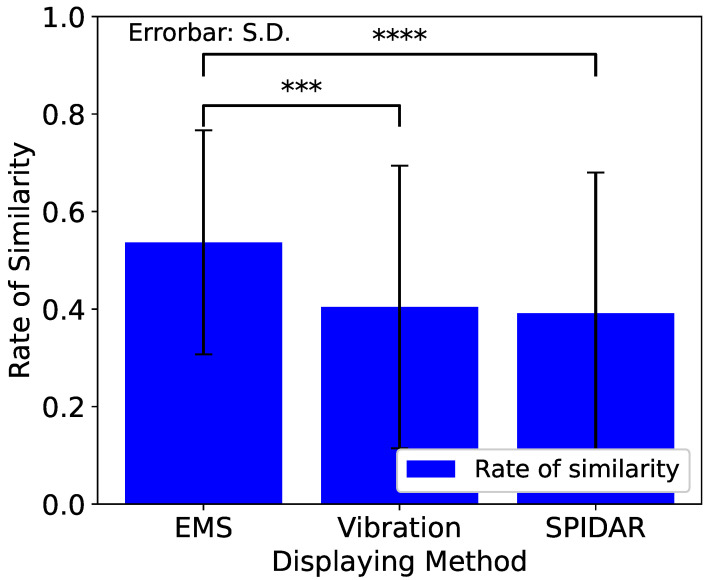
Overall results between methods; the vertical axis is the average similarity for each method. There were statistically significant differences between EMS and vibration and EMS and SPIDAR. *** and **** represent significance levels of 0.05%, and 0.005%.

**Table 1 sensors-23-01796-t001:** Similarity ratios for each method to each real bump. There are three parameters: methods, stimulus intensities, and amplitudes of actual comparative bumps. The first row shows combinations of the methods and stimulus intensities. A numeric value follows after each method name. The numeric value shows the stimulus intensity of each method. The first column shows the amplitudes of the actual bumps. Thus, a total of 27 results are shown.

Real Bump	EMS1	EMS2	EMS3	Vibration1	Vibration2	Vibration3	SPIDAR1	SPIDAR2	SPIDAR3
1 mm	0.518	0.515	0.241	0.440	0.386	0.481	0.543	0.267	0.187
2 mm	0.577	0.569	0.493	0.459	0.467	0.343	0.406	0.458	0.370
3 mm	0.640	0.649	0.630	0.422	0.383	0.256	0.346	0.474	0.471

## Data Availability

The data presented in this study are available on request from the corresponding author.

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
