# Peer review of "Evaluation of Electric Muscle Stimulation Method for Haptic Augmented Reality†"

_sensors, 2023, doi:10.3390/s23041796_

Round 1

Reviewer 1 Report

I want to congratulate the authors for a very well written paper. The research is of a great interest today so other researchers will find valuable information is this paper.

I want to ask the authors to carefully read the paper and correct a few English mistakes they made or  typo errors:

-After that, we presentd the sensation of bumps...

-As a signle trial, after stimulating....

-As we explained in the previous section, we determined to use 14 patterns of waveform and pulse width for input stimuli.

- Therefore, we found that the wave pairs between the difference is over 0.4 ms have some statistically significant differences.

The last two sentences should be rephrased. 

Author Response

Thank you for  your suggestions. I corrected the words as below.

  • -After that, we presentd the sensation of bumps…
    • After that, we presented the sensation of bumps…
    •  
  • -As a signle trial, after stimulating....
    • As a single trial, after stimulating....
  • -As we explained in the previous section, we determined to use 14 patterns of waveform and pulse width for input stimuli.
    • As we explained in the beginning of the section, we determined to use 14 patterns of waveform and pulse width for input stimuli. 
  • - Therefore, we found that the wave pairs between the difference is over 0.4 ms have some statistically significant differences.
    • Therefore, we found that the wave pairs between the difference over 0.4 ms have some statistically significant differences.

Reviewer 2 Report

Thank you for allowing me to read your paper on EMS with Augmented Reality. I think you work is quite interesting and the theme is suitable for the journal. I added some comments that I think will help you improve the final version.

To start with, the opening sentence can be revised: '..there is a lot of research on Augmented Reality (AR) technology' - this is quite ambiguous (e.g. what is 'a lot'?). Next, it is not clear what you mean by 'conventional haptic AR devices are not suitable for interacting with real objects. This is because they prevent direct contact between the user and real objects by their controllers...' I think this can be made sharper and clearer. What do you mean by direct contact?

Line 66 'we proposed a device that users can wear and that can give a force sensation without preventing direct contact with real objects' - the novelty can be clearer as can the research contribution. At the moment the work comes across as development work rather than research  - you could, for instance, add a research question or clarify the actual research contributions. In section 3.1 you conduct experiments, so there is research, but the research purpose is not clear in the Introduction. 

Further it is not clear at present in the introduction how this work differs from other approaches - for example EMS with AR has been used a fair bit over the last few years: one quick example -https://dl.acm.org/doi/abs/10.1145/3173574.3174020

As you have no background section, how this work differs to others must be clearer in the Introduction.

Section 2 is a little unusual. Is it quite short and seems to be providing some background but also some contribution towards the methodology.  It should be revised/merged/expanded ... 

Section 3 - you repeat the word 'examine' - e.g. 'Here, we examined new stimulus waveforms, other than pulse waveforms. At the same time, we examined the appropriate frequency and pulse width'. What do you mean by examine? What is your examination protocol or purpose? 

Further I thought the rest of the paper was fine. Well done.

Author Response

Thank you for your suggestions. We answer each comments as the attachment.

Reviewer 3 Report

The manuscript is devoted to the development of Haptic Augmented Reality methods, in particular, the authors suggest using the Electric Muscle Stimulation Method (EMS) for this purpose. This area of ​​research has been developing rapidly in recent years, so the proposed manuscript is relevant.

The authors proposed a device that users can wear and that can give a force sensation without preventing direct contact with real objects.

For this purpose, the correct stimulus waveforms for EMS other than impulse waveforms were considered, and parameters of these waveforms, such as frequency and pulse width, were tested.

The study was carried out carefully, all experiments are described in detail, the results of the experiments are clearly presented, the results of the experiments and conclusions are correctly argued. 

The manuscript can be published after minor inaccuracies are corrected, in particular, keywords are missing.

Author Response

Thank you for your comment. I appended the keywords as follows;

  • electric muscle stimulation; haptic augmented reality; waveforms

Thank you for your review.